# Association between stringency of lockdown measures and emergency department visits during the COVID-19 pandemic: A Dutch multicentre study

F. Marlijn Booij-Tromp[1], Nicole J. van Groningen[2], Sebastian Vervuurt[3], Juanita A. Haagsma[1,4], Bas de Groot[5], Heleen Lameijer[6], Menno I. Gaakeer[7], Jelmer Alsma[8], Pleunie P. M. Rood[1], Rob J. C. G. Verdonschot[1] *, Marna G. Bouwhuis[1]

1 Emergency Department, Erasmus MC University Medical Center Rotterdam, Rotterdam, The Netherlands, 2 Emergency Department, Franciscus Gasthuis & Vlietland, Rotterdam, The Netherlands, 3 Emergency Department, Jeroen Bosch Hospital, Den Bosch, The Netherlands, 4 Department of Public Health, Erasmus MC University Medical Center Rotterdam, Rotterdam, The Netherlands, 5 Emergency Department, Radboud University Medical Center, Nijmegen, The Netherlands, 6 Emergency Department, Medical Center Leeuwarden, Leeuwarden, The Netherlands, 7 Emergency Department, Admiraal de Ruyter Hospital, Goes, The Netherlands, 8 Department of Internal Medicine, Erasmus MC University Medical Center, Rotterdam, The Netherlands

* r.verdonschot@erasmusmc.nl

## Abstract

### Introduction

The COVID-19 outbreak disrupted regular health care, including the Emergency Department (ED), and resulted in insufficient ICU capacity. Lockdown measures were taken to prevent disease spread and hospital overcrowding. Little is known about the relationship of stringency of lockdown measures on ED utilization.

### Objective

This study aimed to compare the frequency and characteristics of ED visits during the COVID-19 outbreak in 2020 to 2019, and their relation to stringency of lockdown measures.

### Material and methods

A retrospective multicentre study among five Dutch hospitals was performed. The primary outcome was the absolute number of ED visits (year 2018 and 2019 compared to 2020). Secondary outcomes were age, sex, triage category, way of transportation, referral, disposition, and treating medical specialty. The relation between stringency of lockdown measures, measured with the Oxford Stringency Index (OSI) and number and characteristics of ED visits was analysed.

### Results

The total number of ED visits in the five hospitals in 2019 was 165,894, whereas the total number of visits in 2020 was 135,762, which was a decrease of 18.2% (range per hospital:

**Data Availability Statement:** There are legal restrictions concerning data sharing. In the data

sharing agreement with the Netherlands Emergency Evaluation Database (NEED) it was stated (and signed for) that the data would not be provided to a third party, in line with the General Data Protection Regulation (GDPR). Also in the data sharing agreements with the other two hospitals it was stated (and signed for) that the data would not be shared to a third party. This is in line with Erasmus Medical Center policy. The ethical body mandating the restriction of this data is our legal department. Mr. L. Stoffel LLM. – advisor data protection and privacy, and coordinator information safety – can be contacted for this via email address: l.stoffel@erasmusmc.nl.

**Funding:** The author(s) received no specific funding for this work.

**Competing interests:** No authors have competing interests

10.5%-30.7%). The reduction in ED visits was greater during periods of high stringency lockdown measures, as indicated by OSI.

## Conclusion

The number of ED visits in the Netherlands has significantly dropped during the first year of the COVID-19 pandemic, with a clear association between decreasing ED visits and increasing lockdown measures. The OSI could be used as an indicator in the management of ED visits during a future pandemic.

## Introduction

By the end of December 2019, a novel coronavirus was reported in China, now known as SARS-CoV-2. On February 27th the first COVID-19 patient was reported in the Netherlands [1]. The spread in the Netherlands was partly attributed to so-called super spreader events such as winter vacations and mass gatherings at international sports events and Dutch Carnival [2,3]. Many governments implemented country-wide preventative measures against further spread of the virus, including lockdown measures. Lockdown measures and the stringency of these measures varied across countries and over time [4]. After the introduction of the first measures on March 12th 2020, a partial lockdown was announced in the Netherlands starting on March 23th. The lockdown measures included working at home, closure of schools and childcare, no group meetings (<100 people) and traffic restrictions. The impact of lockdown on daily life was immense and was accompanied by great uncertainty, insecurity and fear in the general population [5,6].

During the pandemic, regular healthcare was downsized in order to increase capacity for the care of COVID-19 patients [7]. The increasing number of COVID-19 patients presenting at the Emergency Department (ED) affected the daily routine and work flow. Therefore, adjustments were made such as redirection of medical personnel and hiring more staff [8]. Meanwhile, there were concerns for drastic decreases in ED visits in other patient categories. Early reports from the first months of lockdown in Italy in 2020 showed a reduction in ED visits as high as 68% [9,10]. Similar reductions were reported in other countries such as Germany, the United Kingdom and the United States [11–13]. In these studies, reductions were seen in non-urgent conditions (e.g. mild trauma), but also in life-threatening conditions such as acute coronary syndrome, heart failure and stroke.

Few studies have examined the effect of the pandemic on ED visits in the Netherlands, and focussed mainly on specific patient categories and the first lockdown period [14–19]. To our knowledge, the effects of different stringency levels of lockdown measures on ED visits is not studied.

Therefore, the primary aim of this study was to describe the frequency of ED visits during the first year of the COVID-19 pandemic (2020), compared to the prior two years (2018 and 2019) and to the stringency of lockdown measures. The secondary aims were to describe characteristics of ED visits during 2020, compared to 2019 and to the stringency of lockdown measures.

## Material and methods

### Study design

This retrospective observational multicentre study was conducted in the Netherlands and approved by the Medical Ethics Committee of the Erasmus MC (number 2020–0315). In this study the need for an informed consent was waived due to the retrospective nature of this

study and the use of anonymous data. Data were collected from five Dutch hospitals. Two academic hospitals and 3 non-academic hospital located in different regions of the Netherlands. Two of the participating hospitals were part of the Netherlands Emergency Evaluation Database (NEED) [20]. NEED is a Dutch quality registry for EDs containing clinical data from all ED visits from the participating hospitals. A dataset was only available from two participating hospitals for the study period. See supporting information S1 Table for hospital characteristics.

## Study population and data collection

The frequency and characteristics of ED visits from February 1st, 2019 to January 31st 2020 (year 2019), and February 1st 2020 until January 31st 2021 (COVID-year 2020) were collected. The total number of ED visits of 2019 and 2020 were compared to the reference year 2018. Characteristics of ED visits were; date of the ED visit, age, sex, triage category according to Manchester Triage System (MTS), means of transportation to the ED, referral (e.g. general practitioner, self-referral), destination after ED discharge, and medical specialty (as described below). The Manchester Triage System was used for triage upon arrival at the ED in all five hospitals. Data were extracted from the electronic patient files and the NEED. Data were accessed for research purposes between June 3rd, 2021 until December 30th 2021. Authors had access to pseudonymized data only.

The following main categories for medical specialties were used; surgery (including orthopaedics and plastic surgery), internal medicine (including pulmonology, geriatric medicine, rheumatology, gastroenterology), cardiology (including thoracic surgery), neurology (including neurosurgery), other surgical (including urology, ophthalmology, otorhinolaryngology), paediatrics and others (including gynaecology, intensive care, anaesthesiology, dermatology, psychiatry).

The stringency level of lockdown measures can be defined with the Oxford Stringency Index (OSI). The OSI is an index measuring international government responsiveness during the COVID-19 pandemic and ranges from 1 to 100, with a higher index indicating more and stricter lockdown measures [4].

## Statistical analysis

Descriptive statistics were used to assess the total number of ED visits in 2018, 2019 and 2020. Chi-square tests (categorical variables) and the Mann-Whitney U test and the independent sample T-test (continuous variables) were used to test for differences in characteristics of ED visits between 2019 and 2020. In addition, the number of ED visits (2020) was plotted against the OSI to determine if there was an association between the level of lockdown measures and the number of ED visits.

In order to determine if characteristics of the ED visits (2020) differed across periods with varying OSI, the OSI was divided into five categories: category 1; 0.00–0.99, category 2; 1.00–25.99, category 3; 26.00–50.99, category 4; 51.00–75.99, category 5; 76.00–100 and Chi-square and one-way ANOVA were used to test for differences. Analyses were done per month or per week. Statistical analyses were performed with IBM Statistics 27. A p-value of $<0.05$ was considered statistically significant.

## Results

### Number of ED visits

The total number of ED visits in 2018 and 2019 were 164,290 and 165,894 respectively. For 2020 the total number of visits was 135,762. The decrease in ED visits averaged 18.2%, and ranged by hospital from a low of 10.5% to a high of 30.7% (Table 1).

**Table 1. Annual number of ED visits, total and by hospital.**

| | Hospitals | | | | | Total |
|---|---|---|---|---|---|---|
| | **H1** | **H2** | **H3** | **H4** | **H5** | |
| 2018 | 32,553 | 44,397 | 34,491 | 27,141 | 25,708 | 164,290 |
| 2019 | 33,101 | 45,328 | 35,576 | 26,154 | 25,735 | 165,894 |
| Growth PY (2019 vs 2018) | 1.7% | 2.1% | 3.1% | -3.6% | 0.1% | 1.0% |
| 2020 | 25,389 | 38,901 | 30,303 | 18,126* | 23,043 | 135,762 |
| Growth PY (2020 vs 2019) | -23.3% | -14.2% | -14.8% | -30.7% | -10.5% | -18.2% |

Growth PY = growth relative to prior year (%).H = Hospital.

*Missing data of the last three weeks for H4 (2020).

Fig 1 shows the number of ED visits per week in the study period. The difference in number of ED visits per week between 2019 and 2020 was higher during the first, second and third peak wave of COVID-19 in the Netherlands. The greatest reduction in ED visits was 34%, seen in the first month after the introduction of the first lockdown measures in the Netherlands (week 11–15). The trends in the reduction in ED visits were similar across all hospitals, with slight variations in the magnitude of reductions (Fig 2).

In 2020, the OSI in the Netherlands ranged from 0 (no lock down measures) to 82. Fig 3 shows the number of ED visits compared to the OSI; when OSI increased, the number of ED visits decreased, and vice versa.

## Characteristics of ED visits

The characteristics of the ED visits are shown in Table 2. No relevant differences in age and sex were found between 2019 and 2020. The mean age was 48.1 and 50.8 years, for 2019 and 2020 respectively. In 2019, 52.2% of the patients were male versus 52.8% in 2020. In 2020, a decrease was observed for all age categories with the largest decrease in the category ≤ 17 years (relative difference of -28.4%). In addition, relatively more patients in age category ≥ 70 years were seen in 2020 compared to 2019 (28.6% versus 26.0%; p<0.001).

In 2020, relatively more yellow, orange and red triaged patients, and relatively less green triaged patients by MTS were seen. Also, a greater percentage of patients arrived by ambulance (27.7% in 2019 versus 32.8% in 2020; p<0.001). Furthermore, self-referrals were relatively lower (43.0% in 2019 versus 41.6% in 2020; p<0.001), and referrals by the general physician were higher (50.7% in 2019 versus 51.4% in 2020, p<0.001). The admission rate was higher (35.5% in 2019 versus 41.0% in 2020, p<0.001), but the ICU admission rate was slightly lower (0.8% in 2019 versus 0.7% in 2020, p<0.001). Relatively less surgery and paediatric patients were seen, but relatively more patients for internal medicine, including pulmonology.

Table 3 shows the characteristics of the ED visits by OSI category. As the OSI increased, the mean age increased accordingly. More specifically, the percentage of patients in the age category ≤ 17 years decreased, whereas the patients in the age category ≥ 70 years increased (p<0.001). With an OSI >25, a decrease in the percentage of green triage category and an increase in the percentage of yellow, orange and red triage categories was observed (p<0.001).

Furthermore, as the OSI increased a higher percentage of patients arrived by ambulance and the admission rate increased (both p<0.001). Also a higher OSI was associated with a lower percentage of ED visits for paediatrics. In the highest OSI category (75.00–100.00), most ED visits were patients for internal medicine.

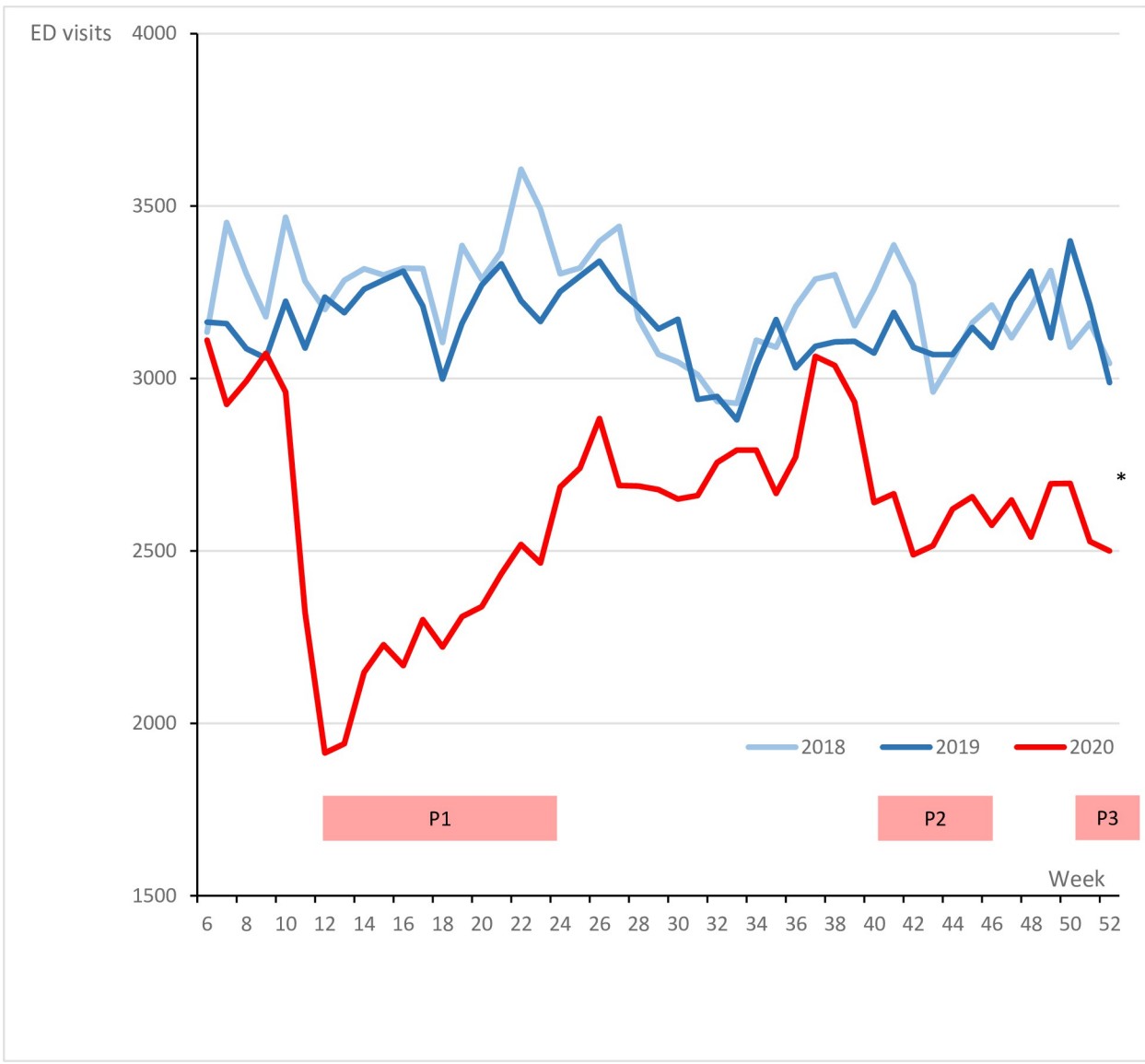

**Fig 1. Total weekly ED visits per year.** P1 = First Covid Peak, P2 = Second Covid Peak, P3 = Third Covid Peak. *Missing data of the last three weeks for H4 (2020).

## Discussion

This study compared frequency and characteristics of ED visits in five Dutch hospitals during the first year of COVID-19 with the pre-pandemic year. To our knowledge, this study it is the first to investigate the association of stringency of lockdown measures on the frequency and characteristics of ED visits as well. The results show a considerable reduction of ED visits in all participating hospitals, although the magnitude of the reduction differed across hospitals and subgroups of patients. Moreover, the number of ED visits appeared to be inversely related to the stringency of lockdown measures. Reductions were mainly seen in patients $\leq$ 17 years. In addition, it appeared that patients were more severely ill compared to 2019, since triage categories were higher, patients presented by ambulance more often, and admission rates were higher.

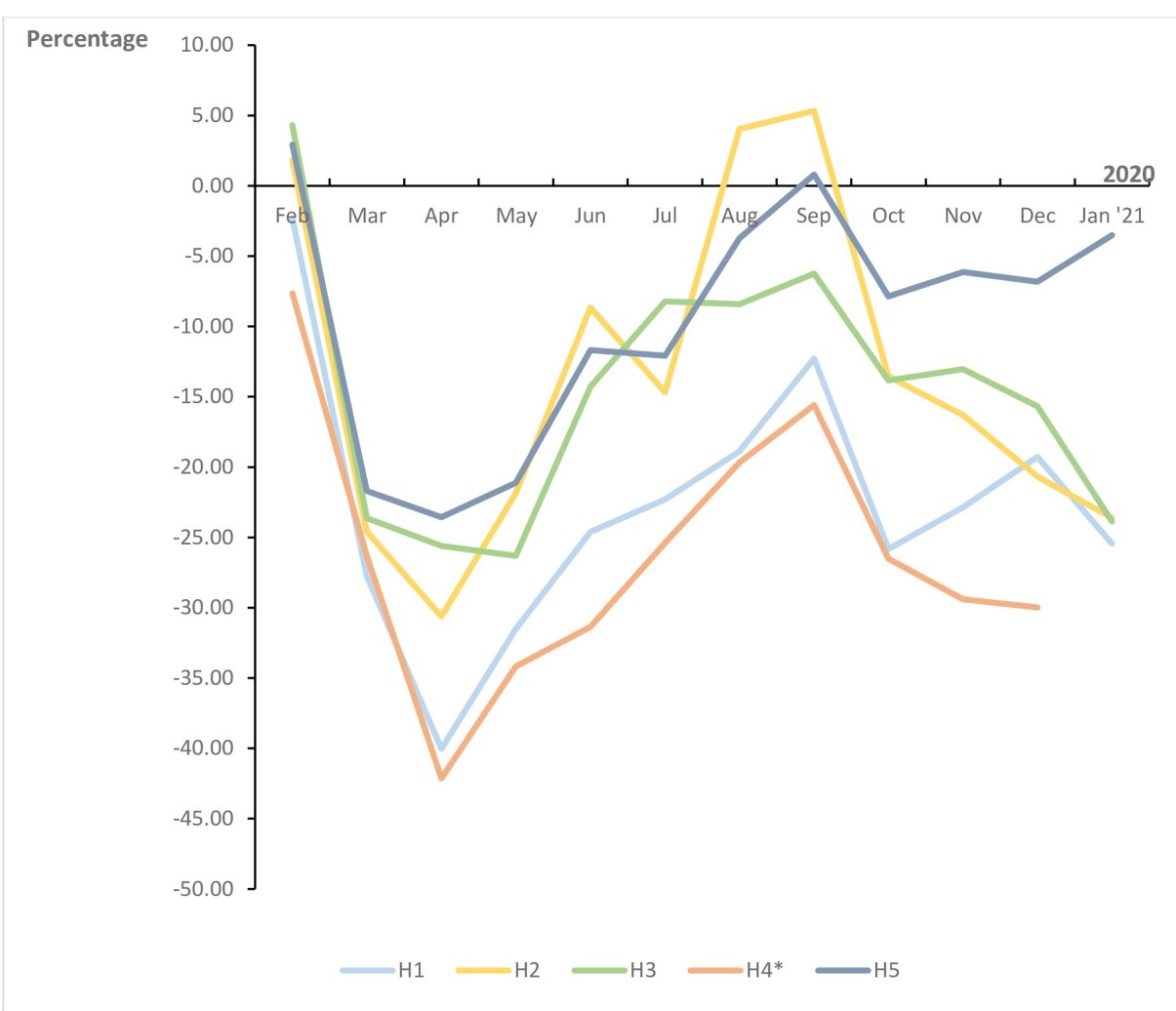

**Fig 2. Difference in ED visits in all participating hospitals in 2020 compared to 2019 in percent per hospital per month.** H = hospital
*Missing data of the last three weeks for H4 (2020).

The current study shows a significant reduction in ED visits of 18.2% in 2020 compared to 2019. Reductions were most pronounced during the first peak wave with a reduction of 34% in the first weeks of lockdown. Other studies from the Netherlands and other countries worldwide showed similar trends [16,19,21–25]. This decline in patient numbers is most likely multifactorial. First, a major reduction in injuries was reported during the pandemic which can be expected as workplace activities, outdoor activities, traffic and nightlife were increasingly restricted [14,15,18]. Second, elective surgeries were postponed which led to fewer patients presenting with postoperative complications [26]. Third, psychological effects played a role as well. One study in the Netherlands reported that patients delayed their visit to the ED in great part due to fear of COVID-19 infection, the wish to not further burden medical professionals, stay at home instructions issued by referring professionals, and the perception of non-urgency of their own complaints compared to COVID-19 patients [27]. The second and third peak wave showed repeated reductions in ED visits, although these were less pronounced than in the first wave. Psychological effects could have worn off as COVID-19 became a part of everyday life and vaccinations were developed, which may have caused decreased adherence to

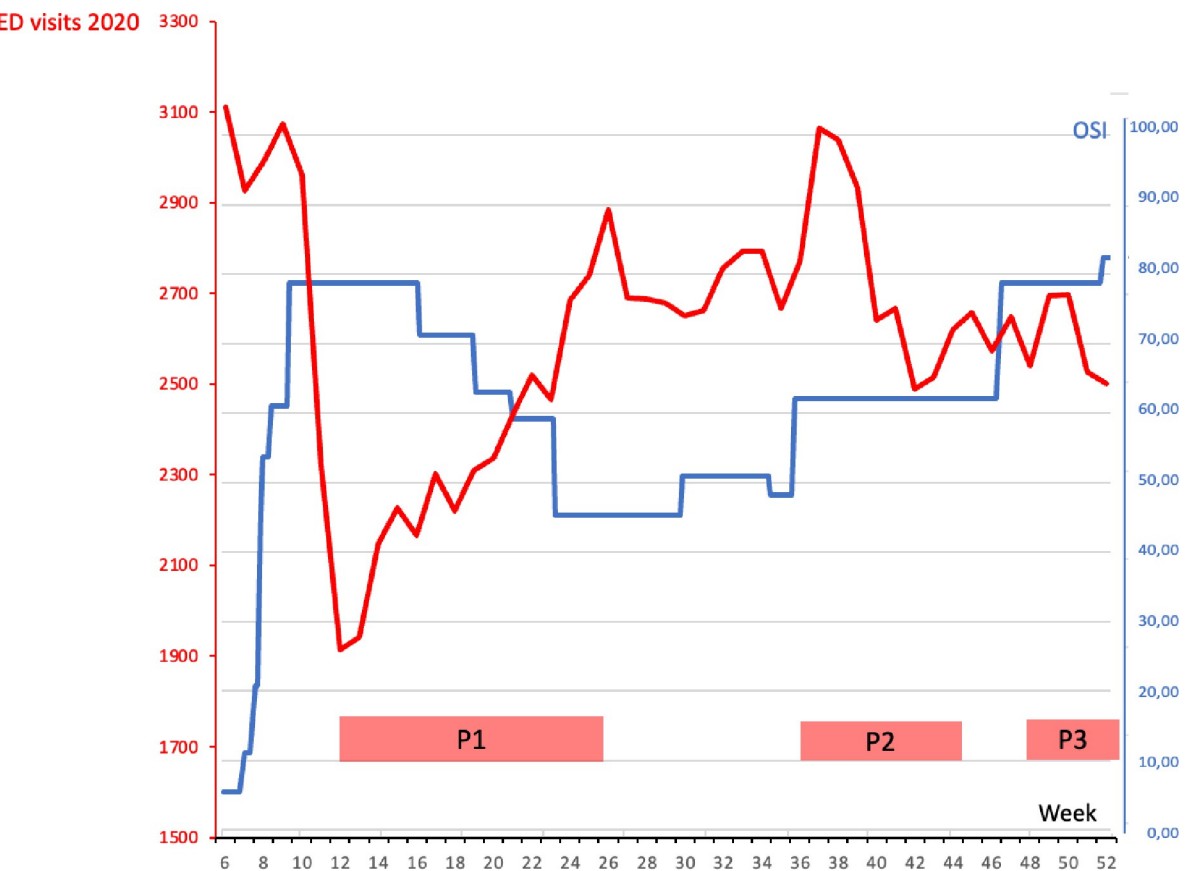

**Fig 3. Weekly total ED visits and Oxford Stringency Index (OSI) in 2020.** OSI = Oxford Stringency Index. *Index for the measurements taken by the government, ranging from 1 (no measurements) to 100 (very strict measurements with a high impact).*

lockdown measures and more ED visits as everyday activities were partly resumed. One study reported that low patient numbers persisted beyond the first and second year of the pandemic, and the authors theorised that this could be permanent as some patients have resorted to other means of healthcare such as telemedicine [24].

The current study shows a relatively higher mean age during the pandemic, with greater absolute and relative reductions in children compared to elderly. Other studies showed similar results [19,21–23,25]. Kruizinga et al. showed that paediatric ED visits in the Netherlands were reduced by 59% in the first half of 2020 with the greatest reductions in infectious diseases [17]. This could be explained by fewer child-to-child interactions and outside activities, decreasing the chances for infection and trauma.

In 2020, the absolute number of elderly patients visiting the ED was lower. This is in line with the study of Dijk et al. [19] In addition to the aforementioned possible reasons, this could be explained to the fact that many elderly died at home or in nursing homes, since COVID-19 infection rates and thereby death rates in nursing homes were high [28,29]. However, relatively more elderly visited the ED, as they were more prone to infection due to higher frailty.

Our study showed a remarkable decrease in ICU admissions and ED mortality in 2020, both in absolute and relative numbers. This is possibly due to the fact that severely ill elderly patients were not admitted to the hospitals during the Covid-19 outbreak anymore and died at home or in nursing homes. And in general, severely ill patients were sent to the ICU expeditiously in order to facilitate the ED.

**Table 2. Characteristics of the ED visits.**

| | | 2019 | | 2020 | | Abs. difference (N) | Rel. difference (%) | Missings | | p-value |
|---|---|---|---|---|---|---|---|---|---|---|
| | | Abs. | Rel. (%) | Abs. | Rel. (%) | | | N | % | |
| **Sex** | Male | 85,636 | 52.2 | 69,526 | 52.7 | -16,110 | -18.8 | 101 | 0.0% | <0.001 |
| | Female | 78,396 | 47.8 | 62,139 | 47.3 | -16,257 | -20.7 | | | |
| **Age** | Mean (years) | 48.1 | | 50.8 | | - | - | none | | <0.001 |
| **Age category** | ≤ 17 years | 25,636 | 15.6 | 18,353 | 13.9 | -7,283 | -28.4 | none | | <0.001 |
| | 18–69 years | 95,792 | 58.4 | 75,693 | 57.5 | -20,099 | -21.0 | | | |
| | ≥ 70 years | 42,658 | 26.0 | 37,666 | 28.6 | -4,992 | -11.7 | | | |
| **MTS category** | Blue | 2,609 | 1.7 | 2,568 | 2.0 | -41 | -1.6 | 17,073 | 5.8% | <0.001 |
| | Green | 52,174 | 34.1 | 40,476 | 32.2 | -11,698 | -22.4 | | | |
| | Yellow | 68,774 | 44.9 | 57,285 | 45.6 | -11,489 | -16.7 | | | |
| | Orange | 27,610 | 18.0 | 23,321 | 18.6 | -4,289 | -15.5 | | | |
| | Red | 2,000 | 1.3 | 1,908 | 1.5 | -92 | -4.6 | | | |
| **Transportation** | Own | 115,905 | 72.3 | 85,674 | 67.2 | -30,231 | -26.1 | 8,068 | 2.7% | <0.001 |
| | Ambulance | 44,305 | 27.7 | 41,846 | 32.8 | -2,459 | -5.6 | | | |
| **Referral** | Self | 70,224 | 43.0 | 54,409 | 41.6 | -15,815 | -22.5 | 1,785 | 0.6% | <0.001 |
| | GP | 82,906 | 50.7 | 67,157 | 51.4 | -15,749 | -19.0 | | | |
| | Specialist | 10,233 | 6.3 | 9,084 | 7.0 | -1,149 | -11.2 | | | |
| **Disposition** | Left on own accord | 151 | 0.1 | 104 | 0.1 | -47 | -31.1 | 1,077 | 0.4% | <0.001 |
| | Home | 71,077 | 43.4 | 54,292 | 41.4 | -16,785 | -23.6 | | | |
| | Admissions | 59,562 | 35.5 | 54,700 | 41.0 | -4,862 | -8.2 | | | |
| | • CU/MCU | 166 | 0.1 | 141 | 0.1 | -25 | -15.1 | | | |
| | • ICU | 1,378 | 0.8 | 855 | 0.7 | -523 | -38.0 | | | |
| | Deceased* | 2,462 | 1.5 | 447 | 0.3 | -2,015 | -81.8 | | | |
| | Transfer | 3,019 | 1.8 | 1,817 | 1.4 | -1,202 | -39.8 | | | |
| | Outpatient Clinic | 22,692 | 13.9 | 18,205 | 13.9 | -4,487 | -19.8 | | | |
| | GP | 4,695 | 2.9 | 1,498 | 1.1 | -3,197 | -68.1 | | | |
| **Medical specialty** | Surgery | 41,560 | 39.6 | 33,157 | 38.5 | -8,403 | -20.2 | 104,803 | 35.4% | <0.001 |
| | Internal Medicine | 32,826 | 31.3 | 28,881 | 33.5 | -3,945 | -12.0 | | | |
| | Cardiology | 4,628 | 4.4 | 3,670 | 4.3 | -958 | -20.7 | | | |
| | Neurology | 11,595 | 11.1 | 9,856 | 11.4 | -1,739 | -15.0 | | | |
| | Other surgical | 5,322 | 5.1 | 4,368 | 5.1 | -954 | -17.9 | | | |
| | Paediatrics | 7,684 | 7.3 | 5,057 | 5.9 | -2,627 | -34.2 | | | |
| | Others^ | 1,260 | 1.2 | 1,131 | 1.3 | -129 | -10.2 | | | |

GP = general practitioner/family doctor, MTS = Manchester Triage System, CCU = Cardiac Care Unit, MCU = Medium Care Unit, ICU = Intensive Care Unit.

* Deceased during ED visit.

^ Gynaecology, Intensive Care, Anesthesiology, Dermatology, Psychiatry.

Patient numbers for all specialties decreased, mostly for surgery and paediatrics and least for internal medicine. Other studies showed an initial sharp decline in severe conditions (acute myocardial infarction, stroke, and sepsis) followed by a recovery phase, with overall numbers remaining low during the first wave [21–23,30]. In other studies, ED visits for trauma followed the same trend [19,21,22,25]. A reduction of trauma in Dutch studies was up to 37% in the first months of the pandemic [14,15,18,19].

As the OSI increased, the mean age increased, triage categories increased (yellow, orange and red), more patients arrived by ambulance, and admission rates were higher. Significantly less paediatric patients and more internal medicine patients visited the ED with increased OSI.

**Table 3. ED visits characteristics by Oxford Stringency Index category.**

| | | Oxford Stringency Index category | | | | | p-value |
|---|---|---|---|---|---|---|---|
| | | 0.00–0.99 | 1.00–24.99 | 25.00–50.99 | 51.00–74.99 | 75.00–100 | |
| **Sex** | Male | 52.2% | 53.5% | 52.8% | 52.9% | 52.7% | 0.019 |
| **Age** | Mean (years) | 48.2 | 48.9 | 49.3 | 50.5 | 52.5 | <0.001 |
| **Age category** | ≤ 17 years | 15.6% | 15.7% | 15.0% | 14.1% | 11.4% | <0.001 |
| | 18–69 years | 58.3% | 56.2% | 57.7% | 57.3% | 57.8% | |
| | ≥ 70 years | 26.1% | 28.1% | 27.3% | 28.6% | 30.7% | |
| **MTS category** | Blue | 1.7% | 2.0% | 2.1% | 2.0% | 2.0% | <0.001 |
| | Green | 34.0% | 33.0% | 34.4% | 31.8% | 29.7% | |
| | Yellow | 44.9% | 45.0% | 44.3% | 45.9% | 47.4% | |
| | Orange | 18.1% | 18.9% | 17.8% | 18.7% | 19.2% | |
| | Red | 1.3% | 1.0% | 1.4% | 1.6% | 1.7% | |
| **Transportation** | Own | 72.3% | 72.0% | 68.1% | 66.8% | 64.1% | <0.001 |
| | Ambulance | 27.7% | 28.0% | 31.9% | 33.2% | 35.9% | |
| **Referral** | Self | 42.9% | 41.7% | 40.8% | 40.7% | 44.8% | <0.001 |
| | GP | 50.8% | 50.6% | 52.6% | 52.2% | 48.4% | |
| | Specialist | 6.3% | 7.7% | 6.7% | 7.1% | 6.9% | |
| **Disposition** | Left on own accord | 0.1% | 0.1% | 0.1% | 0.1% | 0.1% | <0.001 |
| | Home | 43.4% | 43.1% | 43.4% | 41.8% | 37.1% | |
| | Admission | 35.5% | 36.6% | 39.2% | 41.5% | 45.1% | |
| | • CCU/MCU | 0.1% | 0.1% | 0.1% | 0.1% | 0.1% | |
| | • ICU | 0.9% | 0.8% | 0.6% | 0.6% | 0.5% | |
| | Deceased* | 1.5% | 1.2% | 0.2% | 0.2% | 0.2% | |
| | Transfer | 1.8% | 1.8% | 1.5% | 1.2% | 1.2% | |
| | Outpatient clinic | 13.8% | 13.9% | 14.1% | 13.5% | 14.6% | |
| | GP | 2.9% | 2.5% | 0.7% | 1.1% | 0.9% | |
| **Medical specialty** | Surgery | 39.5% | 36.2% | 41.7% | 38.7% | 35.0% | <0.001 |
| | Internal medicine | 31.3% | 34.1% | 30.5% | 33.4% | 38.5% | |
| | Cardiology | 4.4% | 3.8% | 4.2% | 4.3% | 4.1% | |
| | Neurology | 11.1% | 12.2% | 11.3% | 11.5% | 11.2% | |
| | Other surgical | 5.1% | 5.4% | 5.0% | 4.9% | 5.1% | |
| | Paediatrics | 7.3% | 7.3% | 6.0% | 5.8% | 4.8% | |
| | Others^ | 1.2% | 1.1% | 1.3% | 1.3% | 1.4% | |

GP = general practitioner/family doctor, MTS = Manchester Triage System, CCU = Cardiac Care Unit, MCU = Medium Care Unit, ICU = Intensive Care Unit.

* Deceased during ED visit.

^ Gynaecology, Intensive Care, Anesthesiology, Dermatology, Psychiatry.

Given these association between OSI and ED visits, the stringency of lockdown measures might be helpful to predict ED visits and adapt ED policy during a pandemic (e.g. shifting from conventional healthcare to contingency of crisis level of healthcare) [31].

EDs need to be prepared to manage disasters and crises, such as the COVID-19 pandemic, in the short and long term. Sharing the lessons learned during the pandemic is essential [32]. Importantly, an increasing trend in incidence of internal hospital crises and disasters is observed in the Netherlands, requiring emergency managers to adapt their hospital disaster plans and hospital staff to receive regular training with crisis response templates [33,34].

## Strengths and limitations

This is the largest multicentre study describing the impact of the COVID-19 pandemic on all ED visits in Europe. The participating hospitals are representative for the situation in the Netherlands based on geographical location and included both academic and non-academic hospitals. In contrast to other studies it describes ED visits over a full-year period covering multiple peak waves. Furthermore, it is the first study to investigate the association between ED patient and visit characteristics with the OSI.

Unfortunately, one of the participating hospitals could not provide data from the last three weeks of the COVID year. This probably led to some overestimation of the reduction in ED visits, however it only effected the last month. Further, there was a high heterogeneity in documentation of secondary outcomes between different hospital databases, making detailed analysis challenging. Data on mortality in the ED was often incomplete, which made it difficult to interpret. Medical specialties needed to be pooled for reliable analysis, meaning no statistics on specific diseases could be collected. The increased percentage of internal medicine patients was probably caused by COVID-19 patients that were mainly seen by this specialty. During the start of the pandemic no specific diagnosis registration code existed for COVID-19 infections, which made it not possible to assess the number of COVID-19 patients during the study period.

## Recommendations

Lockdown measures have effects on ED visits during a pandemic. Greater uniformity in the registration of patient characteristics enables a more detailed analysis of these effects. Availability of real-time data on ED visits during a pandemic enables policy makers and healthcare managers to assess the impact of measures directly, and to adjust accordingly. An indicator like OSI with a clear association with ED visits and patient characteristics might be beneficial, although further validation is needed.

## Conclusion

The number of ED visits in the Netherlands have significantly dropped during the first year of the COVID-19 pandemic, with a clear association between decreasing ED visits and increasing lockdown measures. Overall, patients seemed more severely ill as triage categories and admission rates were higher, especially with higher OSI. The OSI could be a useful indicator in the management of ED visits and for patient characteristics during a future pandemic.

## Supporting information

**S1 Table. Hospital characteristics.** H = Hospital, ICU = Intensive Care Unit, MCU = Medium Care Unit, CCU = Cardiac Care Unit, GP = general practitioner/family doctor, PCI = Percutaneous Coronary Intervention. (DOCX)

## Author Contributions

**Conceptualization:** Nicole J. van Groningen, Jelmer Alsma, Pleunie P. M. Rood.

**Data curation:** F. Marlijn Booij-Tromp, Juanita A. Haagsma, Bas de Groot, Marna G. Bouwhuis.

**Formal analysis:** F. Marlijn Booij-Tromp, Nicole J. van Groningen, Juanita A. Haagsma, Marna G. Bouwhuis.

**Investigation:** Sebastian Vervuurt.

**Methodology:** F. Marlijn Booij-Tromp, Nicole J. van Groningen, Sebastian Vervuurt, Juanita A. Haagsma, Marna G. Bouwhuis.

**Project administration:** Nicole J. van Groningen.

**Supervision:** Rob J. C. G. Verdonschot, Marna G. Bouwhuis.

**Visualization:** Marna G. Bouwhuis.

**Writing – original draft:** Sebastian Vervuurt, Marna G. Bouwhuis.

**Writing – review & editing:** Bas de Groot, Heleen Lameijer, Menno I. Gaakeer, Jelmer Alsma, Marna G. Bouwhuis.

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
