## [Decision Letter · Decision Letter 0]

12 Feb 2024

PONE-D-23-40778Association between stringency of lockdown measures and Emergency Department visits during the COVID-19 pandemic: a Dutch multicentre studyPLOS ONE

Dear Dr. VERDONSCHOT,

Thank you for submitting your manuscript to PLOS ONE. After careful consideration, we feel that it has merit but does not fully meet PLOS ONE’s publication criteria as it currently stands. Therefore, we invite you to submit a revised version of the manuscript that addresses the points raised during the review process.Please carefully address all the issues highlighted by the referees. In particular, I believe that the public health/disaster medicine side of discussion has much room to be improved. 

We look forward to receiving your revised manuscript.

Kind regards,

Vincenzo Alfano

Academic Editor

PLOS ONE

Journal Requirements:

4. Please include your tables as part of your main manuscript and remove the individual files. Please note that supplementary tables (should remain/ be uploaded) as separate ""supporting information"" files

Reviewers' comments:

Reviewer's Responses to Questions

**Comments to the Author**

1. Is the manuscript technically sound, and do the data support the conclusions?

Reviewer #1: Yes

Reviewer #2: Yes

2. Has the statistical analysis been performed appropriately and rigorously? 

Reviewer #1: Yes

Reviewer #2: I Don't Know

3. Have the authors made all data underlying the findings in their manuscript fully available?

Reviewer #1: No

Reviewer #2: No

4. Is the manuscript presented in an intelligible fashion and written in standard English?

Reviewer #1: No

Reviewer #2: Yes

5. Review Comments to the Author

Reviewer #1: This is a retrospective, multicenter study in 5 Dutch hospitals on the association between ED utilization and the stringency of lockdown measures. Although numerous publications have emerged on ED utilization patterns during the pandemic, this manuscript is interesting because it assesses the association with the OSI index. However, there are some minor concerns and the English language could be improved throughout the manuscript.

GENERAL

The English language could be improved throughout the manuscript, particularly with regards to grammar and style. Some sentences are a little bit odd.

ABSTRACT

- I would refrain from u- sing "usual" healthcare; consider "regular" instead

- You state that little is known about the influence of lockdown on ED visits. This is not true and has been extensively studied. You are right about that little is known about the association of ED utilization and OSI.

- The number of ED visits "have": should be "has"

- I think there is a better wording than "tool" for OSI; maybe indicator is better (also in main article discussion and conclusion)

INTRODUCTION

- COVID-19 is the diseaese that is caused by SARS-CoV-2, so this is not exactly the same.

- Preventative measures do not only represent lockdown measures (such as social distancing, increased hygiene measures, etc), so this part should be rephrased. It would also be good to underline that there is a great variety in lockdown; in the netherlands there never was a lockdown as stringent as in China. So consider to explain a bit more what the (first) lockdown looked like.

- Line 81-82: consider to use the word reference years instead of the two prior years.

METHODS

no specific comments

RESULTS

no specific comments

DISCUSSION

- Good discussion of the present literature

- Line 224: I doubt if vaccination development played a role at that time, because they were not available yet.

- Line 246: consider using ED mortality instead of deceases

Reviewer #2: This study depicts the ED fluxes of patients during the first part of the COVID-19 pandemics, discussing some useful insights for future public health measures.

I think the paper is balanced in its sections; some minor suggestions here:

- comparison 2018+2019 vs 2020 as primary aim but not present in the abstract; maybe just worth mentioning it.

- maybe internal medicine visits preserved an increased trend because only complex cases (usually managed by IM, especially if elderly and if geriatrics not present) bypassed the patients' fear to access EDs.

- relationships between higher ORI and negative consequences can be speculated; e.g. higher ORI could mean higher mortality or excess mortality; similar to the choice of shifting from conventional to contingency to crisis level of healthcare [ DOI: 10.1097/DMP.0b013e31819f1ae2 ; 10.1056/CAT.20.0384 ] so a message for stakeholders: you can choose to apply stricter measures, but expect consequences on population health.

Thank you for letting me review this paper.

6. PLOS authors have the option to publish the peer review history of their article (what does this mean?). If published, this will include your full peer review and any attached files.

Reviewer #1: No

Reviewer #2: No

---

## [Author Response · Author response to Decision Letter 0]

18 Apr 2024

Responses to all the questions and comments of the editor and reviewers can be found in the Response to the Reviewers document as attached. Also find our Statement on Data Sharing attached separately.

---

## [Decision Letter · Decision Letter 1]

2 May 2024

Association between stringency of lockdown measures and Emergency Department visits during the COVID-19 pandemic: a Dutch multicentre study

PONE-D-23-40778R1

Dear Dr. VERDONSCHOT,

We’re pleased to inform you that your manuscript has been judged scientifically suitable for publication and will be formally accepted for publication once it meets all outstanding technical requirements.

Kind regards,

Vincenzo Alfano

Academic Editor

PLOS ONE

Additional Editor Comments (optional):

Reviewers' comments:

Reviewer's Responses to Questions

**Comments to the Author**

1. If the authors have adequately addressed your comments raised in a previous round of review and you feel that this manuscript is now acceptable for publication, you may indicate that here to bypass the “Comments to the Author” section, enter your conflict of interest statement in the “Confidential to Editor” section, and submit your "Accept" recommendation.

Reviewer #1: All comments have been addressed

Reviewer #2: All comments have been addressed

2. Is the manuscript technically sound, and do the data support the conclusions?

Reviewer #1: Yes

Reviewer #2: Yes

3. Has the statistical analysis been performed appropriately and rigorously? 

Reviewer #1: Yes

Reviewer #2: Yes

4. Have the authors made all data underlying the findings in their manuscript fully available?

Reviewer #1: No

Reviewer #2: Yes

5. Is the manuscript presented in an intelligible fashion and written in standard English?

Reviewer #1: Yes

Reviewer #2: Yes

6. Review Comments to the Author

Reviewer #1: Dear authors,

All concerns are now sufficiently addressed.

Great paper. Thank you for letting me review it.

Reviewer #2: (No Response)

7. PLOS authors have the option to publish the peer review history of their article (what does this mean?). If published, this will include your full peer review and any attached files.

Reviewer #1: No

Reviewer #2: No

---

## [Editor Report · Acceptance letter]

10 May 2024

PONE-D-23-40778R1 

PLOS ONE

Dear Dr. VERDONSCHOT, 

I'm pleased to inform you that your manuscript has been deemed suitable for publication in PLOS ONE. Congratulations! Your manuscript is now being handed over to our production team.

Kind regards, 

on behalf of

Dr. Vincenzo Alfano 

Academic Editor

PLOS ONE